# Facile Molecular Weight Determination of Polymer Brushes Grafted from One-Dimensional Diffraction Grating by SI-ATRP Using Reflective Laser System

**DOI:** 10.3390/polym13234270

**Published:** 2021-12-06

**Authors:** Jem-Kun Chen, Feng-Ping Lin, Chi-Jung Chang, Chien-Hsing Lu, Chih-Feng Huang

**Affiliations:** 1Department of Materials and Science Engineering, National Taiwan University of Science and Technology, 43, Sec. 4, Keelung Road, Taipei 106335, Taiwan; jkchen@mail.ntust.edu.tw (J.-K.C.); D10304007@mail.ntust.edu.tw (F.-P.L.); 2R&D Center for Membrane Technology, Chung Yuan Christian University, Taoyuan 32043, Taiwan; 3Department of Chemical Engineering, Feng Chia University, 100, Wenhwa Road, Taichung 40724, Taiwan; changcj@fcu.edu.tw; 4Department of Obstetrics and Gynecology, Taichung Veterans General Hospital, Taichung 40705, Taiwan; 5Institute of Biomedical Sciences, Ph.D. Program in Translational Medicine, and Rong-Hsing Research Center for Translational Medicine, National Chung-Hsing University, Taichung 40227, Taiwan; 6i-Center for Advanced Science and Technology (iCAST), Department of Chemical Engineering, National Chung Hsing University, Taichung 40227, Taiwan

**Keywords:** diffraction grating, PMAA, molecular weight, laser system

## Abstract

Gelatin was immobilized selectively on the amide groups-modified bottom of a trench array of a photoresist template with 2 μm resolution by the ethyl(dimethylaminopropyl) carbodiimide/*N*-hydroxysuccinimide reaction. The gelatin-immobilized line array was brominated to generate a macroinitiator for atom transfer radical polymerization. Poly(methacrylic acid) (PMAA) brushes were grafted from the macroinitiator layer as line arrays of one-dimensional diffraction gratings (DGs) for various grafting polymerization times. A laser beam system was employed to analyze the optical feature with a characteristic diffraction effect of the PMAA DGs at a 45° incident angle along the transverse magnetic and transverse electric polarization. The growth of the PMAA brush lines increased both their heights and widths, leading to a change in the reflective diffraction intensity. The PMAA brushes under various grafting polymerization times were cleaved from the substrate by digestion of gelatin with trypsin, and their molecular weights were obtained by gel permeation chromatography. The change degree of the diffraction intensity varied linearly with the molecular weight of the PMAA brushes over a wide range, from 135 to 1475 kDa, with high correlation coefficients. Molecular weight determination of polymer brushes using the reflective diffraction intensity provides a simple method to monitor their growth in real time without polymer brush cleavage.

## 1. Introduction

Polymer brushes are formed by the tethering of the ends of polymer chains to a surface, leading to various structures [1]. The functional groups of polymer brushes can be manipulated arbitrarily by the monomer species during polymerization for tailoring the physicochemical properties or functionalization of biomolecules [2]. For example, many types of functional groups, such as carboxyl acid, primary amino, epoxy, and hydroxyl groups, can be anchored on the chains of polymer brushes to immobilize biomacromolecules. The flexibility of polymer brushes facilitates their surface modification for several applications, such as biological detection [3], cell affinity substrate [4], and bacterial resistance [5]. Immobilization of biomacromolecules on surfaces using polymer brushes facilitates control of a higher grafting density than that with self-assembled monolayers because of the three-dimensional (3D) topographical structure [6]. There are several approaches for grafting dense polymer brushes from substrates, such as surface-initiated reversible addition−fragmentation chain-transfer polymerization [7], surface-initiated atom transfer radical polymerization (SI-ATRP) [8], and surface-initiated nitroxide-mediated polymerization [9]. Compared to “grafting to” methods, these “grafting from” approaches can design polymer brush architectures with high flexibility and obtain higher grafting density. Among these approaches, SI-ATRP is the most common for controlled radical polymerization owing to the chemical versatility, excellent reliability, and high tolerance to the oxygen in solution. 

Although the molecular weight of a graft chain is an important parameter that affects the properties of the grafted materials for applications in specific fields, the characterization of polymer brushes in real time remains challenging because cleavage of the covalent graft bonds is needed. Cleaving covalent graft bonds from substrates to analyze polymer brushes directly via size exclusion chromatography is a common approach [10]. In previous studies, the covalent graft bonds of polymer brushes were cleaved from silica surfaces by dissolving the silica substrates using hydrofluoric acid (HF) [11]. Moreover, p-toluenesulfonic acid was employed to cleave the covalent graft bonds of polymer brushes grafted from silicon surfaces initiated by an acid-labile linker. However, these approaches can only be applied to HF-dissolvable substrates and HF-insensitive grafts. Compared with nanoparticle substrates, harvesting polymer brushes from large-area substrates for molecular weight determination is still difficult and inconvenient owing to the much smaller amount of sample collection. Furthermore, the free polymers synthesized simultaneously in the bulk solution during polymer brush grafting from a substrate are also employed to determine the molecular weights and polydispersity index (PDI) values [12]. However, experimental data suggest that the molecular weights and PDI values of polymers in bulk solution and on substrates are different [13,14]. 

One-dimensional/two-dimensional (2D) periodic gratings can be fabricated practically by top-down semiconductor processes. Many sensors, such as DNA [15], virus [16], ion [17], and humidity [18] sensors, have been fabricated with periodic gratings using the diffraction effect. A light beam traveling through a periodic grating produces diffracted beams. Correlation between the change in the optical properties of diffracted beams and the geometrical parameters of gratings with the refractive index is established to identify target components [19]. The optical properties of diffracted beams include a shift in the wavelength and spatial distance and the diffraction intensity efficiency of grating patterns, which can be used as indicators to determine species as well as trace targets. Generally, high-resolution patterns of periodic gratings result in large spatial spaces of the diffraction, which can enhance the sensing performance. Recently, photonic crystals with periodic patterns were developed as diffraction-based sensors, which can also cause the light interaction with the periodic structure. The diffraction effect caused by light traveling through regularly periodic patterns is a basic phenomenon, which has been developed in various optics’ engineering [20]. To achieve a desired diffraction effect, periodic gratings are patterned with regular periodic structures on the wavelength scale. When light is incident on such a periodic medium, the reflection or transmission at specific angles exhibits specific spectral properties, different from aperiodic media. Space–time periodic (STP) media can remarkably interact with electromagnetic waves, which has acquired significant scientific attention for application in numerous electromagnetic systems [21,22]. In this study, trench arrays of a photoresist template with a 2 μm resolution are fabricated by a very-large-scale integration process on silicon surfaces. A gelatin backbone is immobilized selectively on the silicon surfaces by an amide groups-modified trench bottom for bromination to serve as the initiator layer in ATRP. SI poly(methacrylic acid) (PMAA) is grafted from the brominated gelatin backbones on the silicon surfaces by ATRP as a line array of diffraction gratings (DGs) [23,24]. PMAA is a type of biocompatible polymer with rich carboxyl groups, which has been extensively used as polymer brushes to bind with proteins for wide applications [25]. The line arrays of the DGs can provide transverse magnetic (TM) and transverse electric (TE) polarization optical modes to vary the diffraction effect. Characterization of the optical properties of the PMAA DGs, including electromagnetic wave propagation and scattering through STP media [26,27], is an interesting topic to correlate with the molecular weights of the brushes. Growth of the PMAA DGs on the surfaces varies the heights and widths of the lines, resulting in a change in the reflective diffraction intensity, which is measured by an in-house-constructed laser system. The change in the diffraction intensity of the PMAA DGs with the grafting polymerization time is used as an indicator to establish the correlation with the molecular weight of the PMAA, obtained by digesting the gelatin backbone with trypsin. Polymer chains stretch and coil in the bulk solution and dry state on the substrates respectively, which may result the difference in measurement of polymer molecular weight. In general, polymers grafted on the surface are stripped from the substrate to measure the exact molecular weight. This strategy offers a convenient approach for monitoring the growth of polymer brushes in real time without destructed cleavage. Although the exact molecular weight of the polymer grafted on the surface is not well-known, the molecular weight measured by our method is expected to be proportional to that molecular weight measured in the bulk solution.

## 2. Experimental Section

### 2.1. Materials

Silicon wafers, purchased from Hitachi, Inc. (Tokyo, Japan), were used after removal of dust particles and organic contaminants. Line arrays were patterned in a region of 1 cm^2^ on one side of the polished silicon wafer. 3-aminopropyltriethoxysilane (3A) and 2-bromo-2-methylpropionyl bromide (2B), used to generate the halogen groups of surface-initiator of ATRP, were purchased from Acros Organics Co. (Geel, Belgium). Methacrylic acid (MAA), triethylamine (TA), 1,1,4,7,7-pentamethyldiethylenetriamin (PMDETA), copper(I) bromide (Cu(I)Br), and copper(II) bromide (CuBr_2_) were obtained from Sigma Aldrich (Saint Louis, MO, USA). The MAA monomer was purified by a series of vacuum distillation processes to remove the inhibitor prior to use. Sigma-Aldrich also supplied the gelatin and trypsin from porcine skin and porcine pancreas (180 TAME units mg^−1^), respectively. *N*-(3-Dimethylaminopropyl)-*N’*-ethylcarbodiimide hydrochloride (EDC), *N*-hydroxysuccinimide (NHS), ammonia, and 3,3′,5,5′-tetramethylbenzidine (TMB) substrate buffer were supplied by Acros Organics Co. Other reagent grade solvents came from Sigma Aldrich and were used without further purification.

### 2.2. Strategy of Molecular Weight Determination with PMAA DGs by Laser Beam System

Figure 1 illustrates the strategy of the molecular weight determination of the PMAA brushes with reflective diffraction intensity using a laser system [28]. (A) A photoresist was coated on Si wafers to pattern as trench arrays of 2 μm resolution by I-line lithography. (B) Oxygen plasma was induced to generate a hydrophilic region in the trench bottom, and subsequently, the photoresist was removed from the surface. (C) The samples were immersed in a 0.5 wt.% 3A aqueous solution for 1 h at 25 °C to assemble selectively on the hydrophilic regions of the wafer substrates. (D) The as-prepared samples were incubated with ethyl(dimethylaminopropyl) carbodiimide (EDC) and *N*-hydroxysuccinimide (NHS) solutions, leading to activation of the amide groups of 3A. Subsequently, a gelatin backbone was immobilized on the 3A-modified samples in phosphate-buffered saline (PBS) solution at 2 mg mL^−1^ for 3 h at 10 °C, denoted as DG-gel. (E) The as-prepared samples were sequentially reacted with 2B for 6 h at room temperature to generate halogen groups on the surface of DG-gel as macroinitiators, denoted as DG-gel-Br. (F) PMAA brushes were grafted from the DG-gel-Br via ATRP using MAA, Cu(I)Br, CuBr_2_, and pentamethyldiethylenetriamine in methanol for 1, 2, 4, 6, 8, and 16 h of grafting polymerization times at 30 °C. These are denoted as DG-g-PMAA1, DG-g-PMAA2, DG-g-PMAA4, DG-g-PMAA6, DG-g-PMAA8, and DG-g-PMAA16, respectively. After various grafting polymerization times, the samples were immersed in a mixture of water and ethanol (1:1, wt.%) for 5 min and then rinsed with double-distilled water at least five times. (G) The DG-g-PMAA samples were evaluated for their diffraction efficiency using an in-house laser beam system. (H) The samples were incubated with trypsin for 30 min to digest the gelatin and release the PMAA brushes from the surfaces for molecular weight analysis by gel permeation chromatography (GPC). Surface components of the samples were analyzed based on the presence and states of the elements by X-ray photoelectron spectroscopy (XPS; Scientific Theta Probe, Waltham, MA, USA). Morphologies of the various DGs were observed via atomic force microscopy (AFM; Veeco Dimension 5000 scanning probe microscope, Plainview, NY, USA). The molecular weights were analyzed via GPC using a VISCOTEK-DM400 instrument (Malvern, UK) and an LR 40 refractive index detector, as described in a previous study [29]. 

### 2.3. Optical Analysis of Reflective STP Grating by a Laser Beam System

STP DG in the reflective mode was analyzed by a home-designed laser beam system. Our DG may be regarded as integration of DG of a PMAA brush and a polished silicon surface with a high reflection efficiency. One may achieve a fully reflective STP DG by combining PMAA brush lines and polished silicon trenches. When electromagnetic waves pass through a line with low resolution at an incident angle, the waves reflecting near an impediment tend to curve around that impediment and reflect out, resulting in wave scattering.

When the impediment features regular repeating patterns, the wave scattering of the electromagnetic waves yields an orderly reflective diffraction phenomenon comprising a series of diffraction orders (m). The reflective diffraction phenomenon is predominately dependent on a comparable optical resolution of grating to the wavelength of the electromagnetic waves traveling through the grating. Two modes of transverse magnetic (TM) and transverse electric (TE) polarization modes occur along directions perpendicular and parallel to the incident laser, respectively. Figure 2a depicts the reflective diffraction from line arrays of STP DGs for TE polarization. When parallel incident waves input the conventional line array of DG for TE polarization, a symmetric diffraction pattern with symmetry in both angles (θ_m_) and amplitudes of the diffracted orders (P_m_) appears. In addition, temporal frequency (ω_0_) of the incident field does not change for a monochromatic input wave, indicating that the diffracted orders possess identical temporal frequency. However, line arrays of DGs possess symmetric profiles, resulting in a reciprocal diffraction reflection response because of restriction of the Lorentz reciprocity theorem [30]. Figure 2b depicts the reflective diffraction from line arrays of STP DGs for TM polarization. When perpendicular incident waves input the STP DG for TM polarization, the nonreciprocity of reflective diffraction occurs. The output waves are asymmetric with the input of the incident wave, indicating no spatial inversion. For TM polarization, the reflective diffraction pattern shows obvious nonreciprocity of the reflective STP DGs. To investigate the angle asymmetry of the reflective STP DG, input incidence of waves under an incident angle (θ_i_) can be employed to analyze the results of output incidence. 

The obvious angle asymmetry, including angles and intensity of the reflective diffracted orders by the DGs, may be observed. The spatial frequency (*K*), spatial periodicity of the STP grating (Λ = 2π/*K*), and the temporal frequency (Ω) can be employed to determine these parameter properties of the diffracted orders according to the momentum conservation law and the energy conservation law [31]. The reflective diffraction angle of output incidence for the diffracted orders can be predicted by the following [27]:(1)$$\sin(\theta_{m})=\frac{\omega_{0}\sin(\theta_{i})+cmK/n_{1}}{\omega_{0}+n\Omega }$$
where *c*, *n*_1_, *m*, and *n* represent the velocity of the light in the vacuum, the refractive index in the air, the numbers of the space, and time harmonics, respectively. The reflective diffraction intensity was recorded as 2D and 3D patterns by a BeamMic system consisting of a laser beam analyzer (Ophir-Spiricon, LLC, North Logan, UT, USA) and BeamMic software.

## 3. Results and Discussion

### 3.1. Surface Characterization of DGs

Figure 1a displays the XPS survey spectra of DG-gel, DG-gel-Br, and DG-g-PMAA16. The XPS spectrum of DG-gel exhibited Si 2p, C 1s, N 1s, and O 1s peaks in the ranges of 99–104, 285–293, 396–403, and 528–535 eV of the silicon wafers, respectively. The surfaces did not present a Br 3d5 peak of the halogen group in the range of 65–73 eV before the bromination reaction. This peak appeared in the XPS spectrum of DG-gel-Br, verifying the presence of the brominated gelatin layer [32]. The signals of the Si 2s, Si 2p, and Br 3d5 peaks disappeared in the DG-g-PMAA16 spectrum. A significant increase in both the carbon to oxygen intensity ratio and nitrogen elemental signal was observed in the DG-g-PMAA16 spectrum due to the coating of the PMAA brushes.

Figure 1b depicts the C 1s high-resolution spectrum of DG-g-PMAA16 with curve fitting to identify the surface bindings. Three characteristic peaks at binding energies of 284.9, 285.6, and 288.5 eV were observed corresponding to the C–C, C–N, and O=C–O bonds, respectively. The binding energies of these surface bindings represent the PMAA grafting on the gelatin, consistent with the values reported in [33]. The characteristic peak of DG-gel-Br at 67.5 eV in the Br 3d high-resolution spectrum disappeared after PMAA grafting (Figure 1c). The intensity of the N 1s high-resolution band of DG-gel-Br was weaker than that of DG-g-PMAA, verifying the PMAA-grafted gelatin (Figure 1d).

Figure 2a shows the 2D/3D AFM topographies and cross-section profiles of the polymer templates for identifying the real topography. The photoresist templates exhibited a 2.4 μm line width and a 1.9 μm trench width with a smooth surface, indicating a slight difference from the designed textures. After the immobilization of gelatin on the bottom of the trenches by the EDC/NHS reaction, clear gelatin lines appeared in the 2D/3D AFM topographies and cross-section profiles. The widths of the gelatin lines matched the trench widths, verifying that the gelatin was grafted on the trench bottom. In the cross-section profiles of DG-gel, the lines had heights of 3.5 nm, suggesting a high regularity of the gelatin immobilization [34,35]. A double-edge structure appeared in each gelatin line, attributed to the edges of the lines readily attaching more biomacromolecules.

As depicted in Figure 3, PMAA is grafted successfully from DG-gel-Br as line arrays of DGs for various grafting polymerization times. All DGs exhibited irregular line surfaces, attributed to the collapse of the PMAA brushes because of the flexibility. The average height of the line patterns of the DG-g-PMAA1 brushes was 145 ± 9 nm, with a distinct interval between the lines. Note that the intervals between the lines suggested a polished Si surface that predominantly reflects the input waves generating the diffraction effect. The irregular textures of the line surfaces were not significantly affected by the diffraction effect. For DG-g-PMAA1, the widths of the bottom regions of the polished silicon wafers remained at 1.9 μm, indicating that the polymer brushes did not extend to the trench bottom regions (Figure 3a). The height of DG-g-PMAA2 increased from 145 ± 9 to 194 ± 13 nm (Figure 3b). The trench bottom regions of the polished silicon wafers were slightly occupied owing to the collapse of the PMAA brushes. As the grafting polymerization time was increased to 4 h, the heights of the PMAA lines continuously increased to 237 ± 18 nm (Figure 3c). Extension of the PMAA brushes to the bottom regions was clearly observed, suggesting that their growth affected the diffraction effect. After PMAA grafting for 6 and 8 h, the heights were 262 ± 18 and 284 ± 20 nm respectively, which did not exhibit any significant changes. In comparison, the occupation degrees of the trench bottom regions by PMAA were remarkably enhanced for DG-g-PMAA6 and DG-g-PMAA8, indicating that the growth of the polymer chains predominantly extended to the trench bottom regions, instead of the heights (Figure 3d,e). The height of DG-g-PMAA16 reached 377 ± 30 nm without clear trench bottom regions (Figure 3f). Note that the trench region of the polished silicon surface predominantly affected the diffraction effect. The change in the widths of the trench bottom regions with PMAA growth was expected to correlate with the reflective diffraction intensity.

### 3.2. Measurement of Diffraction Efficiency of PMAA DGs

Figure 4a presents the measurement of the reflective diffraction of the PMAA DGs. The change in the morphologies of the PMAA DGs reduces the regularity of the periodic structures, weakening the reflective diffraction intensity. The 2D and 3D diffractive patterns present the uniformity and intensity of the laser beam reflected from the DGs.

Figure 4b,c display the reflective diffraction patterns of the laser beam from DG-g-PMAA16 for the TM and TE polarization, respectively. The green spots represent the diffracted orders reflected by the laser beam from the DGs for the TM and TE polarization. For the TE polarization, the laser beam travels parallel through the grating as well as through multi-slits, resulting in a diffraction phenomenon that is perpendicular to the incident laser beam (Figure 4b). Equation (1) can be used to calculate the angles of the diffracted orders at a particular incident angle, where the diffraction orders satisfying |sin (θm)| < 1 are called “propagating” orders. These evanescent orders play important roles in several surface-enhanced grating properties and are considered in the equation, suggesting that the intensities reduce exponentially with the distance from the DGs. Diffracted orders at a distance less than a few wavelengths from the DGs were selected to analyze the diffraction properties. For the TM polarization, the reflective diffraction spots were ordered with various diffraction angles along the input laser beam owing to nonreciprocity (Figure 4c). Incidence of the laser beam at 45° was considered to investigate the angle asymmetry of the reflective DGs. (Table 1) A full angle asymmetry, including the angles and intensities of the reflective diffracted orders, of the DGs was clearly observed for the TM polarization. Note that the fuzzy diffractive multi-spots of PMAA DG suggest that the direction of the input laser beam was not well-aligned along the TM polarization. For the TE polarization, the diffraction spots were ordered along a horizontal direction perpendicular to the direction of the input wave (Figure 2a). Thus, the diffraction angles at diffracted orders (m) from −3 to +3 were almost identical to those from −3 to +3, indicating the reciprocity of the diffraction.

For the TM polarization, the diffraction angles at m from −3 to −1 were significantly different from those from +1 to +3, indicating the angle asymmetry of the diffraction due to the nonreciprocity (Figure 2b).

Note that the m ranging from −1 to +3 waves was presented as propagating diffracted orders, whereas m ≤ −2 and m ≥ +4 were diffracted as evanescent orders. Compared to the diffraction angles for the TM polarization, those for the TE polarization exhibited better angle symmetry. Therefore, it was more convenient to analyze the diffraction due to the larger interval among the diffractive spots for the TM polarization. Figure 5a,b display the 2D and 3D patterns of DG-gel and DG-gel-Br collected from the reflection at a 45° incident angle for the TM polarization.

The reflective intensity of the laser beam from the polished silicon wafer without any patterns was 62.3 Mcnts. The reflective diffraction intensity of DG-gel at m = 0 was ca. 47.2 Mcnts, indicating that the diffraction effect was significant through the line array. Since the trench bottom regions of DG-gel did not change significantly after the bromination reaction, the diffraction intensity reduced slightly from 47.2 to 46.8 Mcnts for DG-gel-Br. Figure 6a–f show the 2D and 3D diffraction patterns of PMAA DGs for various grafting polymerization times for the TM polarization at m = 0. The diffraction intensity decreased gradually as the grafting polymerization time changed from 1 to 16 h, indicating that the polymer brushes substantially suppressed the diffraction intensity. The results exhibited a high correlation between the grafting polymerization time and the diffraction intensity. To analyze the inverse proportional dependence of the diffraction intensity on the grafting polymerization times, the change in the diffraction intensity with the grafting polymerization time was defined as E_d_ = (D_0_−D_i_)/D_0_ to evaluate the correlation and the linear range. Here, D_0_ and D_i_ represent the diffraction intensities of DG-gel-Br and PMAA DGs respectively, for various grafting polymerization times. Figure 7a displays the heights of the PMAA brushes and the E_d_ values as functions of the grafting polymerization time. The heights of the PMAA brushes increased rapidly after 1 h of grafting polymerization time, whereas the height increase rate of PMAA significantly retarded after 2 h of grafting polymerization time. The results suggest that the growth of the PMAA brushes not only contributed to the height increase but also to the side extension in the trench bottom regions because of the collapse of the polymer brushes.

The trench bottom regions of the PMAA DGs became gradually occupied with the growth of the polymer chains, modifying the diffraction intensity. We observed an approximately linear increase in E_d_ as the grafting polymerization time increased from 1 to 16 h. To correlate the dependence of the reflective diffraction intensity on the PMAA molecular weight, 300 μL of trypsin was dropped on 1 cm^2^ PMAA DGs at room temperature to cover the surfaces for 30 min to digest the gelatin layer. Subsequently, these chips were rinsed for 10 min using the PBS solution. The cleaved PMAA chains were collected from the rinsed solution by dialysis to measure the molecular weight by GPC. Figure 7b shows the dependence of the molecular weight of a PMAA brush for various grafting polymerization times on the E_d_ value. Linear relationships of Mn and Mw corresponding to the E_d_ values were observed. The linear regression equations were Mn (kDa) = 23.8 (E_d_ (%)) + 5.07 and Mn (kDa) = 33.6 (E_d_ (%)) + 5.8, with correlation coefficients of 0.9984 and 0.9988 respectively, with the PDI values ranging from 1.32 to 1.61. The high correlation coefficients indicate that the E_d_ values can be used to rapidly monitor the growth of polymer brushes in real time without polymer cleavage. The intensity of the reflected laser beam from the polished silicon wafer with a pattern-free PMAA layer did not change significantly, suggesting that the molecular weight was predominated by the diffraction effect, instead of reflection. Our proposed approach could rapidly determine the molecular weights of polymer brushes with high accuracy and reliability, which can provide a rapid real-time indicator for monitoring their growth.

## 4. Conclusions

PMAA brushes were grafted from a brominated gelatin line array under various grafting polymerization times as DGs via ATRP. The growth of the PMAA brushes not only contributed to the height of the lines but also their widths. Concurrently, side extension of the polymer brushes with their growth reduced the widths of the trench bottom regions, which predominantly determined the reflective diffraction intensity. A laser beam system was designed to analyze the reflective diffraction intensity of the line arrays of the DGs for TM and TE polarization with 2D and 3D diffraction patterns. The diffracted spots were ordered along directions perpendicular and parallel to the input wave for the TE and TM polarization, respectively. A line array with a 2 μm designed resolution for the TM polarization was efficient for analyzing the diffraction intensity because of the ability of the trench width to extend the polymer brushes with their growth. The growth of the polymer brush DGs changed the geometrical structure corresponding to the diffraction intensity with grafting polymerization. The diffraction intensity change ratio (E_d_) along the TM polarization at m = 0 can be regarded as an indicator for monitoring the growth of the polymer brushes. A correlation between the molecular weight and the E_d_ values was established, which presented a wide linear range and high correlation coefficients. In comparison with free radical grafting polymerization, SI-ATRP could form the regular pattern of polymer brushes to exhibit good accuracy and reliability for real-time analysis for molecular weight determination by our proposed approach. Furthermore, heights of the grafted polymers below 50 nm made it difficult to establish the calibration curve. The height of the grafted polymers over 100 nm was better to achieve higher accuracy in the measurement of molecular weight. Since the crystalline structure of polymers may interfere with the laser travel through the polymer, molecular weight determination for polymer brushes with a crystalline structure by our proposed approach needs to be further investigated in the future work. This strategy offers a convenient approach for monitoring the growth of polymer brushes in real time without destructed cleavage for molecular weight determination.

## Data Availability

Not applicable.

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
