# Peer review of "Facile Molecular Weight Determination of Polymer Brushes Grafted from One-Dimensional Diffraction Grating by SI-ATRP Using Reflective Laser System"

_polymers, 2021, doi:10.3390/polym13234270_

Round 1
Reviewer 1 Report
Lin et al. report the PMAA brushes grafting from a brominated gelatin line array anchors into the surface of silicon substrates. They interestingly realized a relation between the gratings numbers/spaces and the reflective diffraction intensity. They were able to design a laser beam system to analyze the reflective diffraction intensity of the line arrays of the PMAA DGs to finally use it in the molecular weight determination. The manuscript is informative, well-organized, and English is acceptable. The materials’ surface was well characterized and interpreted as well. This manuscript is recommended to be considered for publications in polymers after tacking out these minor comments:
- The authors should explain the reasons behind the difference in polymers molecular weight in bulk solution and on substrates as they mentioned in the introduction.
- There are many typos, please revise carefully.
- I wonder if the author can apply the same method on more polymers for molecular weight estimation and for precise comparison supporting their method.
- How did you determine the negative values of m, or you just suppose due to symmetry?
- Did the author try to check crystallinity of these polymers using XRD? I wonder if there is any relation with the crystallinity diffraction patterns and gratings spaces from the PMAA DGs.
- Which word is the better description, grafting or gratings? The author should revise throughout the manuscript.
Author Response
Reviewer #1:
Lin et al. report the PMAA brushes grafting from a brominated gelatin line array anchors into the surface of silicon substrates. They interestingly realized a relation between the gratings numbers/spaces and the reflective diffraction intensity. They were able to design a laser beam system to analyze the reflective diffraction intensity of the line arrays of the PMAA DGs to finally use it in the molecular weight determination. The manuscript is informative, well-organized, and English is acceptable. The materials’ surface was well characterized and interpreted as well. This manuscript is recommended to be considered for publications in polymers after tacking out these minor comments:
1.The authors should explain the reasons behind the difference in polymers molecular weight in bulk solution and on substrates as they mentioned in the introduction.
â–²Our reply: Polymer chains stretch and coil in the bulk solution and dry state on the substrates, respectively, which may result the difference in measurement of polymer molecular weigh. In general, polymers grafted on the surface are stripped from the substrate to measure the exact molecular weight. This strategy offers a convenient approach for monitoring the growth of polymer brushes in real time without destructed cleavage. Although the exact molecular weight of the polymer grafted on the surface is not well-known, the molecular weight measured by our method is expected to be proportional to that molecular weight measured in the bulk solution. We have explained the reasons behind the difference in polymers molecular weight in bulk solution and on substrates as they mentioned in the introduction.
2.There are many typos, please revise carefully.
â–²Our reply: The typos have been corrected.
3.I wonder if the author can apply the same method on more polymers for molecular weight estimation and for precise comparison supporting their method.
â–²Our reply: We have applied the same method on other polymers for molecular weight estimation and for precise comparison supporting their method. However, most heights of grafted polymer were below 50 nm for over 24 h of polymerization time. Heights of the grafted polymer below 50 nm is difficult to establish the accurate calibration curve. The height of the grafted polymers over 100 nm is able to achieve the higher accuracy in measurement of molecular weight. We are going to optimize the grafting polymerization of other polymers now. More polymers for molecular weight estimation should be included in our future work.
4.How did you determine the negative values of m, or you just suppose due to symmetry?
â–²Our reply: There are diffraction spots from -3 to +3 of m. In comparison with other spots, spot location at m = 0 is readily predictable to set up the light pathway for measurement, we only measured the diffraction intensity at m = 0. To measure the intensity at other diffraction spots, diffraction angles and distance have to be coordinated carefully. The light pathway of laser beam system needs to be reset according to the angle and distance for several times until a stable value is obtained. We select the location at m = 0 for measurement to popularize our technology. Establishment of new light pathway to measure other diffraction spots will be included in our future work.
5.Did the author try to check crystallinity of these polymers using XRD? I wonder if there is any relation with the crystallinity diffraction patterns and gratings spaces from the PMAA DGs.
â–²Our reply: PMAA is a kind of amorphous polymer without crystalline structure. In comparison with free polymer, tethered polymer possesses lower freedom to form the crystalline structure. Because the crystalline structure of polymer may interfere with the laser travel through the polymer, we speculate that diffraction intensity of crystalline polymer grating may be varied. We will graft polymers with crystalline structure to investigate the relation between crystallinity and diffraction behavior in our future work.
6.Which word is the better description, grafting or gratings? The author should revise throughout the manuscript.
â–²Our reply: Grafting and grating represent a polymerization and a patterned device, respectively. They are totally different words. We have checked the words throughout the manuscript.
Reviewer 2 Report
The paper is very interesting and the results are carefully interpreted. I believe that after a few minor adjustments it can be published.
- p. 4: How did you rinse the sample after PMMA grafting to get rid of free particles?
- p. 6: According to Fig.1a the carbon to oxygen intensity ratio increased after PMAA grafting, not decreased.
- p.10: Correct Figure 4.
- p. 15: Pull the two graphs apart in Figure 7, because the y axis descriptions are very close to each other and the legend of the graphs is unclear.
- p. 15: It should be Ed or Ed?
Author Response
Reviewer #2:
The paper is very interesting and the results are carefully interpreted. I believe that after a few minor adjustments it can be published.
1.p. 4: How did you rinse the sample after PMMA grafting to get rid of free particles?
â–²Our reply: After various grafting polymerization times, the samples were immersed in a mixture of water and ethanol (1:1, wt %) for 5 min and then rinsed with doubly distilled water at least five times to get rid of free particles. We have included the paragraph in the manuscript.
2.p. 6: According to Fig.1a the carbon to oxygen intensity ratio increased after PMAA grafting, not decreased.
â–²Our reply: It is a typo. We have replaced the item "decrease" with "increase".
3.p.10: Correct Figure 4.
â–²Our reply: We have checked and corrected Figure 4.
4.p. 15: Pull the two graphs apart in Figure 7, because the y axis descriptions are very close to each other and the legend of the graphs is unclear.
â–²Our reply: The two graphs in Figure 7 have been pulled apart.
5.p. 15: It should be Ed or Ed?
â–²Our reply: We have uniformed the symbol of Ed throughout the manuscript.
Reviewer 3 Report
The number of self-citations is very large
The paper uses an ingenious strategy to determine the relationship between laser diffraction and grafted polymer brushes.
In the XPS analysis, the C1s high resolution deconvoluted spectra of the other samples should be also represented. The O1s high resolution spectra should be also reported and discussed. Moreover, as for DG-g-PMAA in Figure 1c the other configuration should be reported in the graph even if no peaks were detected.
In table 1 it is not clear how much independent repetition (and calculated SD) were performed for each PMMA containing sample.
The author in their conclusion should elaborate more about the repeatability, limitations, and extension of this method in different polymer brushes systems.
The number of self-citations is very large, please reduce its impact in the references.
Author Response
Reviewer #3:
The paper uses an ingenious strategy to determine the relationship between laser diffraction and grafted polymer brushes.
In the XPS analysis, the C1s high resolution deconvoluted spectra of the other samples should be also represented. The O1s high resolution spectra should be also reported and discussed. Moreover, as for DG-g-PMAA in Figure 1c the other configuration should be reported in the graph even if no peaks were detected.
â–²Our reply: The C1s and O 1s high resolution spectra of the other samples have been included in another manuscript that is submitted to other journals, which may not be able to include in this manuscript. This manuscript is focused on the methodology of measurement of molecular weight of polymer with our laser beam system. Another manuscript is focused on the synthesis of the grafting polymerization.
In table 1 it is not clear how much independent repetition (and calculated SD) were performed for each PMMA containing sample.
â–²Our reply: Each measurement was repeated independently three times. We have included the independent repetition and SD in the Table 1.
The author in their conclusion should elaborate more about the repeatability, limitations, and extension of this method in different polymer brushes systems.
â–²Our reply: In comparison with free radical grafting polymerization, SI-ATRP could form the regular pattern of polymer brushes to exhibit good accuracy and reliability for real-time analysis for molecular weight determination by our proposed approach. Furthermore, heights of the grafted polymer below 50 nm is difficult to establish the calibration curve. The height of the grafted polymers over 100 nm is better to achieve the higher accuracy in measurement of molecular weight. Because crystalline structure of polymer may interfere with the laser travel through the polymer, molecular weight determination for polymer brushes with crystalline structure by our proposed approach needs to be further investigated in the future work. This strategy offers a convenient approach for monitoring the growth of polymer brushes in real time without destructed cleavage for molecular weight determination. We have elaborated more about the repeatability, limitations, and extension of this method in different polymer brushes systems.
The number of self-citations is very large, please reduce its impact in the references.
â–²Our reply: We have reduced the number of self-citations.
Round 2
Reviewer 3 Report
The author correction and explanation are enough to recommend the paper publication